# Hard Carbon Embedded with FeSiAl Flakes for Improved Microwave Absorption Properties

**DOI:** 10.3390/ma15176068

**Published:** 2022-09-01

**Authors:** Xiaogang Sun, Yi Liu, Daitao Kuang, Jun Lu, Junyi Yang, Xiaomin Peng, Anru Wu

**Affiliations:** 1Hunan Institute of Engineering, College of Mechanical Engineering, Xiangtan 411104, China; 2Hunan Engineering Research Center of New Energy Vehicle Lightweight, Xiangtan 411104, China; 3School of Computational Science and Electronics, Hunan Institute of Engineering, Xiangtan 411104, China

**Keywords:** FeSiAl, hard carbon, layered structure, microwave absorption, impedance matching

## Abstract

Carbon-based composites have been proven to be strong candidates for microwave absorbers in recent years. However, as an important member, magnetic hard carbon (HC)-based composites have rarely been studied in the field of microwave absorption. In this study, HC embedded with FeSiAl (FeSiAl@HC) was synthesized by pyrolyzing a mixture of FeSiAl flakes and phenolic resin (PR). The as-synthesized HC-FeSiAl exhibited a layered structure, and the detailed microstructures were modified by changing the mass ratio of FeSiAl flakes and PR. Thus, the as-synthesized HC-FeSiAl exhibited tunable magnetic properties, wealthy functional groups, excellent thermal stability, and enhanced microwave absorption properties. The optimal minimum reflection loss is lower up to −36.1 dB, and the effective absorption bandwidth is wider up to 11.7 GHz. These results indicated that HC-FeSiAl should be a strong candidate for practical applications of microwave absorption, which may provide new insight into the synthesis of magnetic HC-based composites.

## 1. Introduction

With the coming of 5G communication and the artificial intelligence era, the increase in electromagnetic (EM) pollution has become a serious problem for human health and electrical equipment reliability [1]. Hence, microwave absorption materials (MAMs) have been widely studied in many countries. The ideal MAMs should not only be strong, wide, light, and thin but also meet the technical requirements of environmental friendliness, low-cost approaches, and plentiful availability [2]. Carbon-based materials [1,3,4,5], ceramics [6,7,8,9,10], and magnetic alloy powders [3,11,12,13,14] represent typical potential microwave absorption materials.

Hard carbon (HC), with a nanolayered microstructure and many defect sites as the centers of polarization losses, has exhibited outstanding absorption abilities by encapsulating FeCoNi nanoparticles [3] or being embedded with ZnO nanoparticles [15]. However, the nanoparticle aggregate properties and high cost hinder their practical application. Hence, suitable micron-sized particles are challenging to obtain. FeSiAl powder material is a typical magnetic alloy microwave absorber due to its excellent magnetic properties, high thermal stability, and low cost [11,16,17,18,19,20,21,22,23]. To reduce the Snoek limit and modify the permeability and permittivity of FeSiAl powders, balling, surface oxidation, and coatings are widely used to enhance the microwave absorption properties of FeSiAl materials [11,16,18,19,21,24]. The combination of HC and FeSiAl should be an important strategy to obtain improved microwave absorption properties. However, there are limited reports on the HC-FeSiAl composite as a MAM. 

In this work, magnetic HC-FeSiAl composites, which were composed of layered HC and FeSiAl flakes, were synthesized by pyrolyzing mixtures of FeSiAl flakes and phenolic resin (PR). The as-synthesized HC-FeSiAl exhibited a layered structure, and the detailed microstructures were modified by changing the mass ratio of FeSiAl flakes and PR. The micromorphology, microstructure, static magnetic property, thermal stability, and microwave absorption properties of HC-FeSiAl composites were systematically studied to obtain new insight into the design of related magnetic HC-based composites microwave absorbers.

## 2. Materials and Methods

Commercially available micron-sized FeSiAl flakes (Si 9.8 wt%, Al 5.6 wt%, and Fe 84.6 wt%, Jiangsu Baona Electromagnetic New Material Co., Ltd., Zhenjiang, China) with a purity of 99% were used in this study. The other materials included phenolic resin (Shanghai Macklin Biochemical Co., Ltd., Shanghai, China, 99.8%) and ethanol (Shanghai Titan Chemical Co., Shanghai, China, 99.7%). FeSiAl powders and phenolic resin (mass ratio of 1:1, labeled as S1) were stirred with a magnetic stirrer for 0.5 h with 8 mL of ethanol. The mixture in the quartz boat was placed in vacuum for 0.5 h, then annealed at 800 °C for 3 h in a flowing hydrogen atmosphere, and finally cooled down to room temperature. The synthesized hard carbon embedded with FeSiAl composites (FeSiAl@HC) displayed dark products with a glossy surface. For comparison, FeSiAl flakes to phenolic resin mass ratios of 1:2, 1:3, and 1;4, labeled as S2, S3, and S4, respectively, were prepared following the same procedure. The raw FeSiAl flakes were named as S0. The synthesis of FeSiAl@HC composites is outlined in Figure 1.

The crystal structures of FeSiAl and FeSiAl@HC composites were determined by powder X-ray diffraction (XRD) using Cu-Kα radiation (Bruker D8). Micromorphology and chemical composition were determined on a scanning electron microscope (SEM, Hitachi SU3500) and using energy-dispersive spectroscopy (EDS, IXRF Systems). The static magnetic measurements were examined using a vibrating sample magnetometer (VSM, Lake Shore 7404). The chemical composites and states were tested by Fourier transform infrared spectrometer (FTIR; Nicolet iS 5 FT-IR). The thermal stability was tested by thermogravimetric and differential scanning calorimetry (TG-DSC) with a NETZSCH STA 449 F3. The electromagnetic parameters from 2.0 to 18.0 GHz were tested by a vector Network Analyzer (Agilent N5224B). The powders were uniformly dispersed in paraffin in a mass ratio of 2:1. The powders-paraffin composites were toroidal with an inner diameter of 3.04 mm, an outer diameter of 7.0 mm, and a thickness of 2.0 mm. 

## 3. Results and Discussion

### 3.1. Micromorphology and Microstructure

The structure and morphology of the samples were examined by SEM. As shown in Figure 2a, FeSiAl flakes could be observed with a thickness of ∼1 μm and diameters of 20–90 μm. After being embedded into HC, these flakes exhibited a layered structure (Figure 2b). The high-resolution image of S1 (Figure 2b) indicates that the orientation of FeSiAl flakes was largely irregular. We also found a space between the FeSiAl flakes. This is possibly because a higher content of FeSiAl flakes facilitates dividing the HC into species. As the content of HC increased, the space enlarged, as shown in Figure 2d. This is because HC easily forms a solid entirety, leading to the further separation of the FeSiAl flakes. As the content of HC increased, FeSiAl flakes were completely embedded into large solid HC. Single FeSiAl flakes were difficult to observe. As shown in Figure 2e,f, the surfaces of S3 and S4 were almost HC, with several pieces of FeSiAl flakes. These results suggested the formation of FeSiAl@HC composites, which was also confirmed by the analyses of TEM images (Appendix A). 

Figure 3 depicts the XRD patterns of the FeSiAl and FeSiAl@HC samples. For S0 to S4 in Figure 3, the diffraction peaks located at 27.1°, 31.4°, 45.0°, 65.5°, and 83.0° can be categorized as the (111), (200), (220), (400), and (422) planes of FeSiAl (PDF# 45-1206), respectively. Neither the two diffraction peaks located at approximately 23.7° and 43.6° of hard carbon [15], nor the peak at approximately 26° for graphitized [18] or amorphous [25] can be found in Figure 3. The reason may come from two aspects: first, hard carbon belongs to non-graphitizable carbon with a very low degree of crystallinity; second, the micron-sized flaked-FeSiAl devastates the crystallization process of hard carbon (as shown in Figure 2). 

Figure 4 displays the FTIR spectra of the FeSiAl@HC samples. The FTIR analyses showed that four samples (S1–S4) had similar spectra. The strongest absorption peaks at around 3441 cm^−1^ for all five samples corresponded to the hydroxyl stretching band of the O-H [15,26]. The second strongest absorption peaks at approximately 1096 cm^−1^ for S1 to S4 samples were assigned to the stretching band of C-O. The other bands were associated with C-H, C=C, and C-H vibrations [15,26]. The functional groups in FeSiAl@HC can act as the polarization centers, resulting in the enhancement in the dielectric loss [15].

Figure 5 presents the magnetic hysteresis loops of the FeSiAl and FeSiAl@HC samples at room temperature. The five samples exhibited the typical S-shape characteristic of soft magnetic alloys. The values of saturation magnetization (Mr) of the five samples were 116.0, 64.4, 58.1, 50.2, and 40.3 emu/g for S0–S4, respecitvely. They correspond well to the contents of the flake-shaped FeSiAl alloy powders (Table 1). A similar phenomenon was found for other FeSiAl composites [24]. 

Figure 6 presents the TG-DSC curves of the samples heated in the ambient atmosphere with a heating rate of 10 °C min^−1^. For S0, as shown in the inset in Figure 6, a slight weight loss of 1.2% from 20 to 38 °C was observed due to evaporation of the physiosorbed water, and then a small weight gain occurred because of the oxidation of FeSiAl. This process released a large amount of heat, leading to the exothermic peak at 100 °C. As for S1 to S4, the FeSiAl flakes were protected by the HC. This caused the oxidation process to shift to a higher temperature. We observed that a steady weight loss below 5% from 30 to 400 °C due to the slow oxidation of HC was accompanied by an increase in heat. A sharp weight loss at the higher temperature was observed, which we attributed to the ignition of HC and FeSiAl, accompanied by strong exothermic peaks in the corresponding temperature ranges. The weight losses for S1, S2, S3, and S4 were 57%, 62%, 65%, and 72%, respectively, suggesting an increased content of HC. These results suggested that the as-fabricated FeSiAl@HC possesses excellent thermal stability and enables applications in the microwave absorption field.

### 3.2. Microwave Absorption Properties

To research the absorbing performance of the FeSiAl and FeSiAl@HC composites, the reflection loss (*RL*) versus frequency and thickness were calculated using Formulas (1) and (2) following the transmission line theory [27,28]: *Z_in_* = *Z*_0_ (*μ_r_*/*ε_r_*) ^1/2^
*tanh*[*j* (2*πfd*/*c*)(*μ_r_*/*ε_r_*)^1/2^](1)
*RL* = 20*log*/*(Z_in_* − *Z*_0_)/(*Z_in_* + *Z*_0_)/(2)

Here, *Z_in_* and *Z*_0_ represent the input impedance at the interface of the absorber and the impedance of free space, respectively; *µ_r_* and *ε_r_* are the parameters of complex permeability and complex permittivity, respectively; *f* is the microwave frequency; *d* is the thickness of the absorber; and *c* is the velocity of light in free space, respectively. Generally, *RL* values ≤−10 dB correspond to a more than 90% attenuation ability of the incident microwave. Therefore, the effective absorption bandwidth (the frequency range for *RL* ≤ −10 dB) is abbreviated as Δ*f* [29].

Figure 7 reveals the *R**L* contours and *RL*-2D curves of five samples at 2–18 GHz. As shown in Figure 7a,b, S0 (pure FeSiAl flakes) presented an *RL_min_* of −32.16 dB (16.72 GHz) and a Δ*f* of 2 GHz with a 5 mm thickness. However, with a large value of *RL*, the narrow Δ*f* and large thickness of the S0 sample limit its practical application. The *RL_min_* values with located frequencies and corresponding optimal thicknesses were −22.28 dB (12.72 GHz) at 2.5 mm for the S1 sample, −21.21 dB (11.20 GHz) at 2.5 mm for the S2 sample, −36.10 dB (11.12 GHz) at 3 mm for the S3 sample, and −18.7 dB (17.28 GHz) at 2 mm for the S4 sample. Moreover, a broader Δ*f* was achieved by the FeSiAl@HC composites (2.56 GHz for S1, 12.32 GHz for S2, 11.76 GHz for S3, and 6.32 GHz for S4) by calculating the different distances of the dotted line across *RL* ≤ −10 dB. The microwave absorption parameters of FeSiAl and FeSiAl@HC composites are listed in Table 2, with the other results of the FeSiAl carbon composites. For comparison, the FeSiAl@HC composites exhibited enhanced microwave attenuation capacity, a broadened absorption bandwidth, reduced thickness, lightened weight, lower cost, and facile fabrication. Thus, FeSiAl@HC composites show excellent potential as microwave absorption materials.

Figure 8 displays the real permittivity (*ε*′), imaginary permittivity (*ε*″), dielectric loss tangents (*tanδ_ε_*) from 2.0 to 18.0 GHz, and Cole–Cole semicircles (*ε*″/*ε*′) for FeSiAl and FeSiAl@HC composites. As shown in Figure 8a, the values of *ε*′ of S0, S1, and S2 fluctuated by approximately 7, 6, and 8 from 2 to 18 GHz, except having a small trough at approximately 11.5, 12, and 13.5 GHz, respectively. However, S3 and S4 showed large decreases from 11.5 at 2 GHz to 5 at 9.5 GHz, and from 14 at 2 GHz to 5.5 at 11.5 GHz, and then fluctuated in the range of 5–8 to 18 GHz, respectively. As shown in Figure 8b, the *ε*″ values of S0, S1, and S2 were almost constant until approximately 10 GHz, and then a small peak appeared, and finally gradually increased to 18 GHz. However, just like the trend in *ε*′, the *ε*″ values of S3 and S4 were large and showed large changes. The *ε*″ of S3 slightly decreased from 8 to 6 until 8 GHz, then sharply decreased to 1 at about 13 GHz, and then slowly increased to 3.5 until 18 GHz, while the *ε*″ of S4 slowly dropped from 11.5 to 3.5 throughout the testing period. The tan*δ_ε_* of the five samples followed a similar change trend to *ε*″, except that there was a large peak from the S3 sample and two large peaks from the S4 sample, as shown in Figure 8c. 

The large difference in the dielectric parameters of the five samples followed the mathematical expression (*ε*″ = 1/(2*πε*_0_*ρf*)) based on the free electron theory (*ρ* is resistivity). Generally, a low resistivity enlarges the *ε*″ value. In this work, different contents of hard carbon with the FeSiAl composite resulted in different resistivities. In addition, the dielectric loss mechanism can be understood by the Debye theory following Formula (3) [3,28]:(3)$$\left(\varepsilon^{\prime }-\frac{\varepsilon_{s}+\varepsilon_{\infty }}{2}\right)+{\left(\varepsilon^{\prime \prime }\right)}^{2}={\left(\frac{\varepsilon_{s}-\varepsilon_{\infty }}{2}\right)}^{2},$$
where *ε_s_* and *ε*_∞_ are the static permittivity and the relative permittivity at infinite frequency, respectively. Therefore, the curve of *ε*′ vs. *ε*″ is a semicircle named the Cole–Cole semicircle. A larger radius and more semicircles indicate a stronger Debye dipolar and a stronger dielectric loss [30,31]. The large Deby dipolar values of S3 and S4 were from the high content of hard carbon. There were many defects and functional groups (Figure 4) in the hard carbon matrix. They were the center of the dipoles and thus increased the Deby relaxation [15]. Another reason may have been the optimized layered structure of the S3 sample. 

Figure 9 depicts the real permeability (*μ*′), imaginary permeability (*μ*″), magnetic loss tangent (*tanδ_μ_*), and the impedance matching parameter (*Z_in_*/*Z*_0_) for the FeSiAl and FeSiAl@HC composites. As shown in Figure 9a, the values of *μ*′ for five samples were very close and rapidly decreased from approximately 1.4 to about 0.85 as the frequency increased during the testing period, but a peak occurred at approximately 14 GHz for S1. The changing trends in the values of *μ*″ and *tanδ_μ_* for the five samples were similar to those of *μ*′, except for a broad peak from the S3 sample, as shown in in Figure 9b, c. The real permeability (*μ*′), imaginary permeability (*μ*″), and magnetic loss tangent (*tanδ_μ_*) of FeSiAl are affected by the carbon matrix, similar to FeSiAl/graphite [16,18] and carbon-encapsulated FeSiAl hybrid fakes [11]. Figure 9d displays the dependences of the normalized input impedance *Z_in_*/*Z*_0_ in the five samples. The value of *Z_in_*/*Z*_0_ from the S3 sample approached one across a wider frequency range than that of the other four samples. A value of *Z_in_*/*Z*_0_ close to one indicates the best impedance matching and thus excellent microwave absorption properties. Therefore, the excessively complex permittivity of the S4 sample is generally undesirable.

## 4. Conclusions

Magnetic HC-FeSiAl composites, which were composed of layered HC and FeSiAl flakes, were synthesized by pyrolyzing mixtures of FeSiAl flakes and phenolic resin (PR). Their specific microstructures were tuned by changing the mass ratio of FeSiAl flakes and PR. The results of comparative tests indicated that the as-synthesized HC-FeSiAl composites exhibit tunable magnetic properties, wealthy functional groups, excellent thermal stability, and enhanced microwave absorption property. The optimal *RL_min_* is lower up to −36.1 dB, and the corresponding Δ*f* is wider up to 11.7 GHz. The enhanced microwave absorption property can be attributed to the optimal microstructure, which thus improves the relaxation process, tunable dielectric and magnetic loss properties, and optimal impedance matching property of FeSiAl@HC composites. This study proves that magnetic HC-based composites should be strong candidates as microwave absorbers, providing new insight into the design of related composites. The as-fabricated FeSiAl@HC composites are expected to be a material used in applications of steeling technology and microwave shielding. In addition, the facile method proposed in this study is expected to be applied in the fabrication of other magnetic HC-based composites.

## Figures and Tables

**Figure 1 materials-15-06068-f001:**
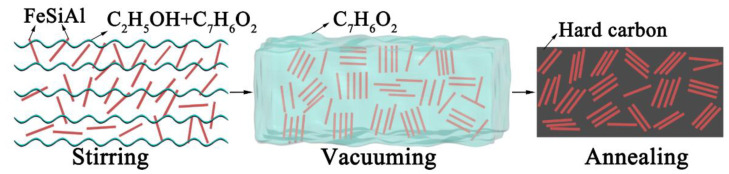
Schematic of the fabrication of the FeSiAl@HC composites.

**Figure 2 materials-15-06068-f002:**
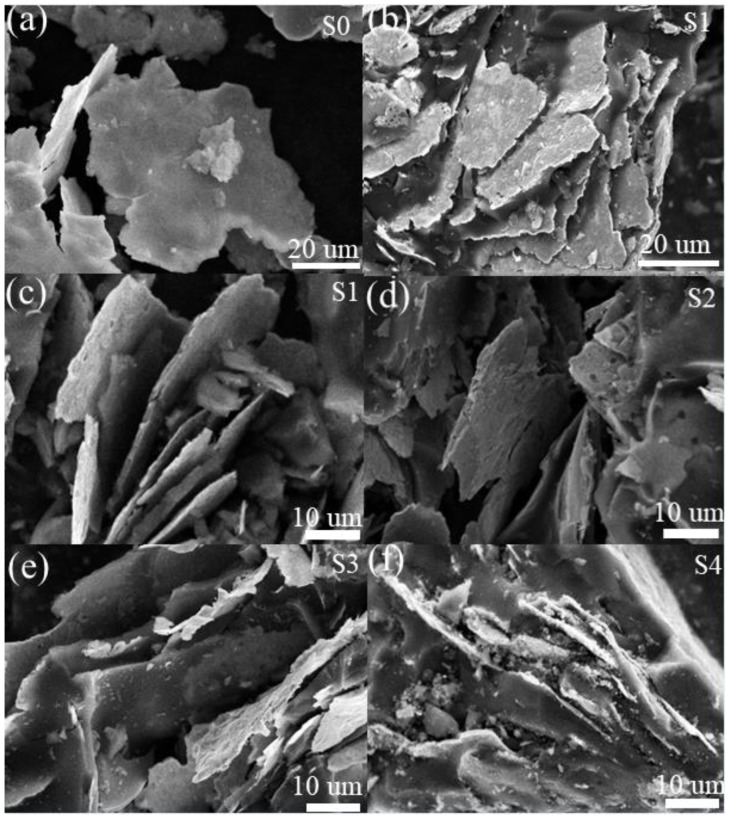
(**a**) SEM image of FeSiAl; SEM images of FeSiAl@HC: (**b**,**c**) S1, (**d**) S2, (**e**) S3, and (**f**) S4.

**Figure 3 materials-15-06068-f003:**
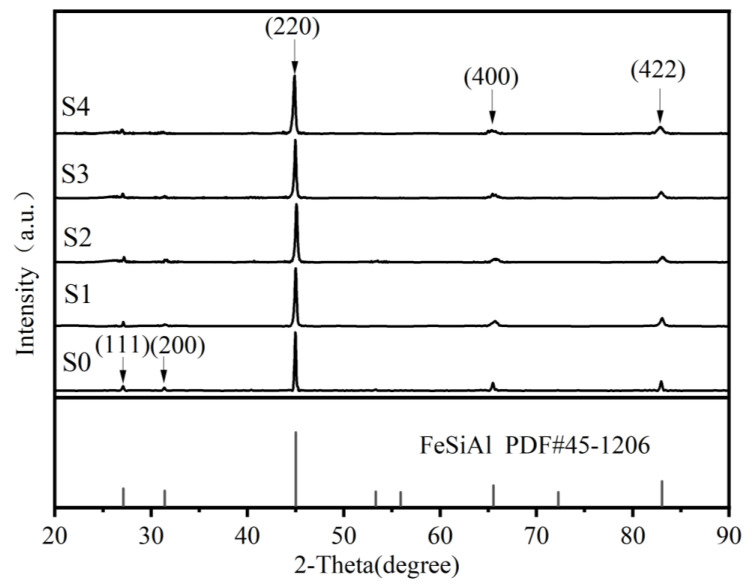
XRD patterns of the FeSiAl and FeSiAl@HC S0−S4 samples.

**Figure 4 materials-15-06068-f004:**
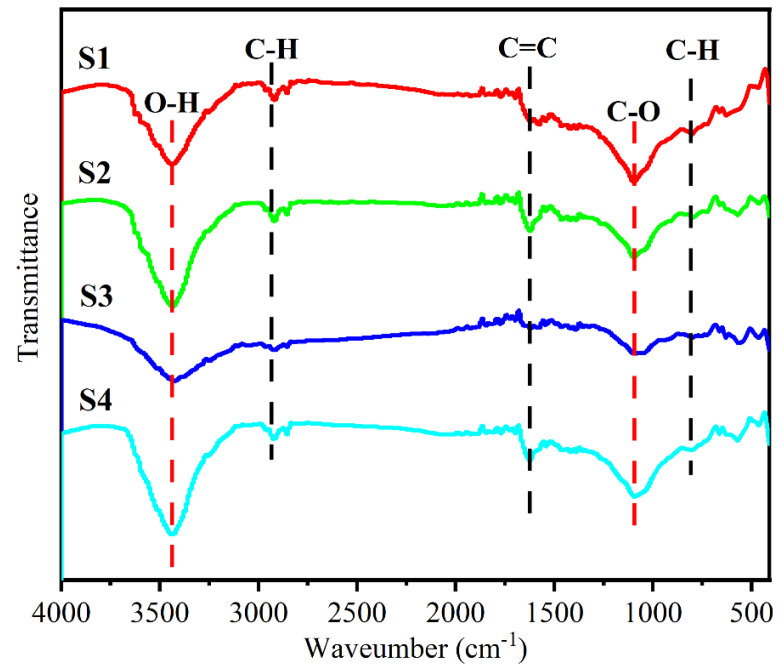
FTIR spectra of the FeSiAl@HC S1−S4 samples.

**Figure 5 materials-15-06068-f005:**
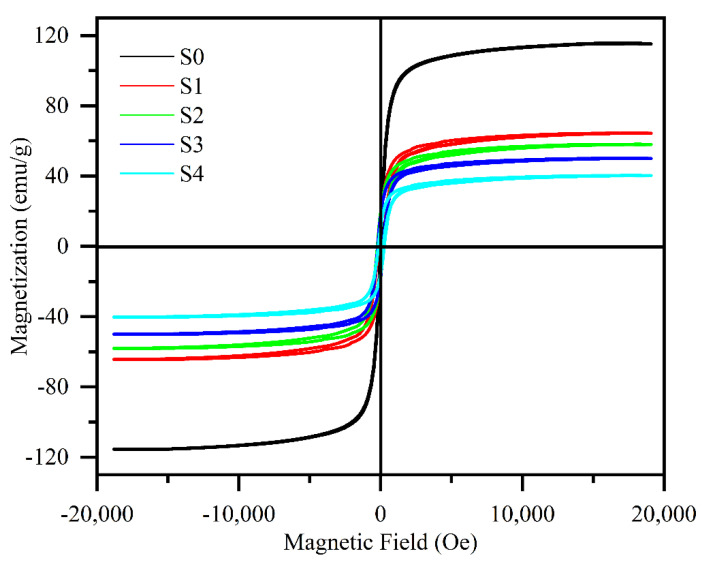
VSM patterns of the FeSiAl and FeSiAl@HC S0−S5 samples.

**Figure 6 materials-15-06068-f006:**
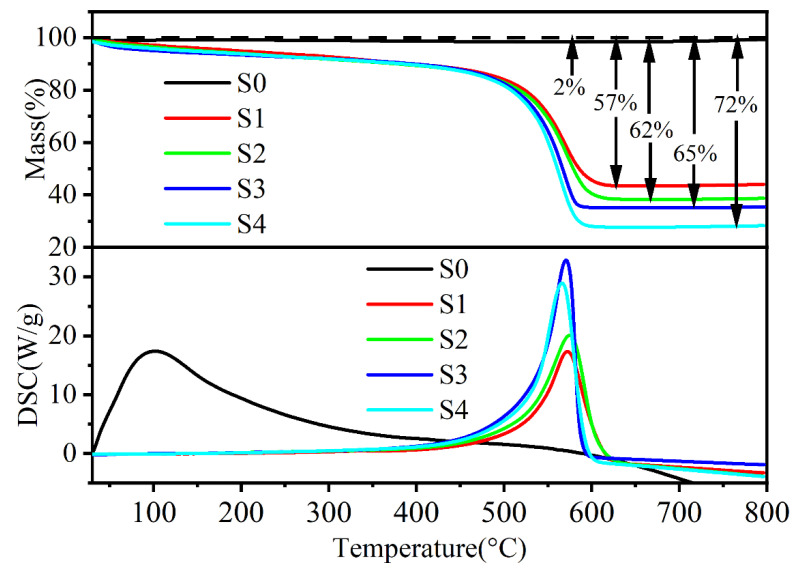
TG−DSC curves of the FeSiAl and FeSiAl@HC S0−S5 samples; the inset shows the enlarged TG−DSC curves from 30 to 80 °C.

**Figure 7 materials-15-06068-f007:**
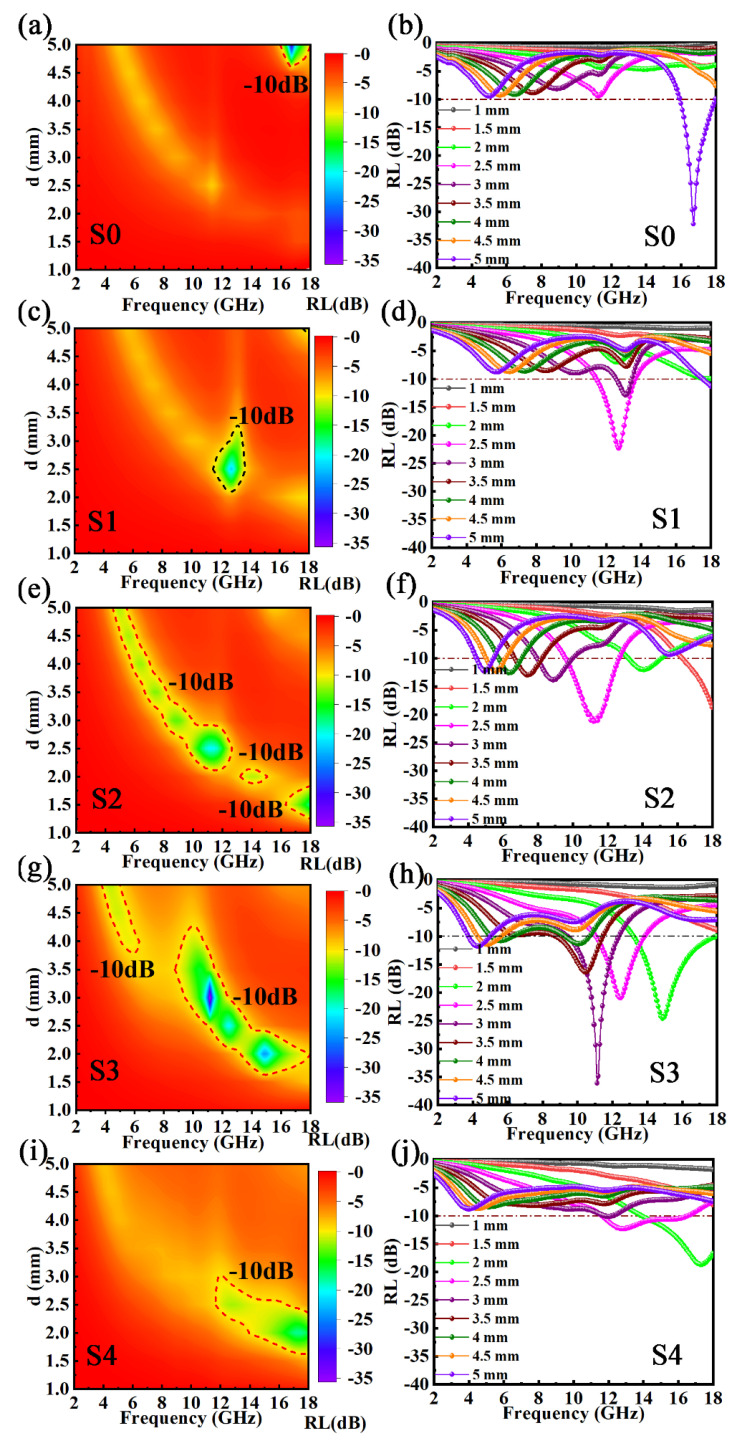
*RL* contours and *RL*−2D curves of the FeSiAl and FeSiAl@HC samples: (**a**,**b**) S0, (**c**,**d**) S1, (**e**,**f**) S2, (**g**,**h**) S3, and (**i**,**j**) S4.

**Figure 8 materials-15-06068-f008:**
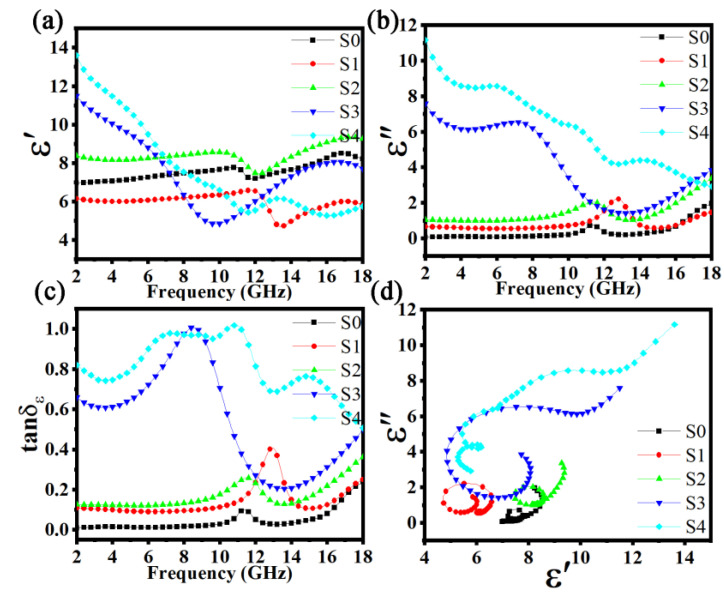
Dielectric parameters of the FeSiAl and FeSiAl@HC samples: (**a**) *ε*′ − *f*, (b) *ε*′ − *f*, (**c**) *tanδ_ε_* − *f*, and (**d**) Cole–Cole semicircles.

**Figure 9 materials-15-06068-f009:**
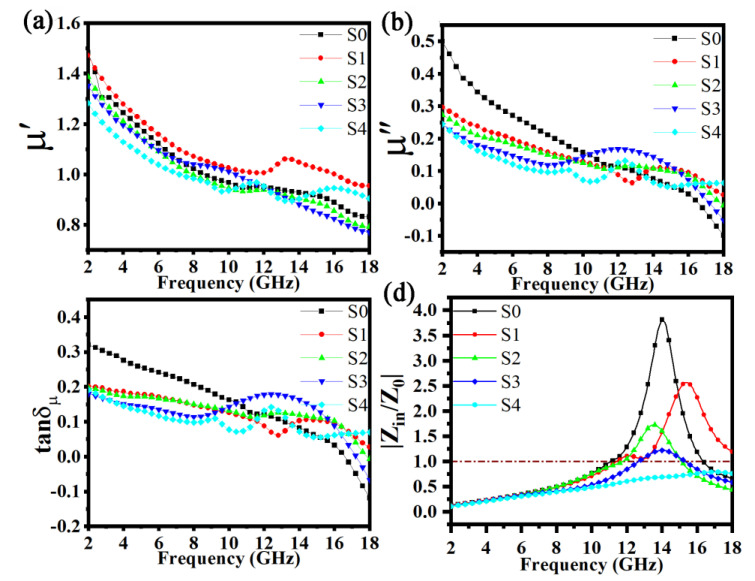
Frequency dependence of the FeSiAl and FeSiAl@HC samples: (**a**) *μ*′ − *f*, (**b**) *μ*″ − *f*, (**c**) *tanδ_μ_* − *f*, and (**d**) *Z_in_*/*Z*_0_ − *f*.

**Table 1 materials-15-06068-t001:** Calculated mass ratio values of saturation magnetization (Mr) of the FeSiAl and FeSiAl@HC samples.

	S0	S1	S2	S3	S4
FeSiAl (wt%)	100	56.2	38.7	30.0	24.3
HC (wt%)	0	43.8	61.7	70.0	75.7
Mr (emu/g)	116.0	64.4	58.1	50.2	40.3

**Table 2 materials-15-06068-t002:** Microwave absorption parameters of related absorbers.

Absorber	Content (wt%)	Carbon (wt%)	RL_min_ (dB)	t_m_ (mm)	Δ*f* (GHz) (RL ≤ −10 dB)	Reference
S0	25	0	−32.16	5	2	This work
S1	25	43.8	−22.28	2.5	2.56	This work
S2	25	61.7	−21.21	2.5	12.32	This work
S3	25	70	−36.10	3	11.7	This work
S4	25	75.7	−18.70	2	6.32	This work
FeSiAl/graphite	40	10	−14	4	6.5	[18]
FeSiAl/graphite	40	20	−23	3.5	13.3	[18]
FeSiAl@C			−15.68	4	2	[11]

## Data Availability

Not applicable.

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
