# Peer review of "Hard Carbon Embedded with FeSiAl Flakes for Improved Microwave Absorption Properties"

_materials, 2022, doi:10.3390/ma15176068_

Round 1

Reviewer 1 Report

In the review of the manuscript titled: Hard carbon embedded with FeSiAl flakes for improved microwave absorption properties. The authors have shown the good description and manuscript is prepared very well. I would like to see this article publish but after some minor modification as follow;

1-I would like the authors to add some description of magnetic and FTIR in the abstract to make the manuscript more attractive.

2-In introduction portion there is very few information present regarding FeSiAl material. Kindly discuss some literature review of recent articles.

3-In Schematic diagram for the fabrication of the FeSiAl@HC composites, the writing within the diagram is very blurred.please update the diagram again.

4-Micromorphology and microstructure on the page 2, line 81, formatting is not same.

5-In SEM diagram, why S1 is mentioned twice with morphological changes. Please explain it.

6- Why peaks of XRD are getting dim with S0 to S4?

7-Morphology is reported as nanoflakes but it seems like sheets and nanoparticles at S4, please suggest appropriate name for morphology.

8- In application point of view please add a line in conclusion in the end for the utilization of the material in devices.

Author Response

Dear Reviewer,

Thank you very much for your valuable comment.

Kind regards

Reviewer 2 Report

The manuscript written by Sun et al. titled ‘Hard carbon embedded with FeSiAl flakes for improved    2 microwave absorption properties’ describes the syntheses of hard carbon layers with FeSiAl embedded for microwave absorption. 

The following points needs to be addressed 

  1. EDX analysis should be performed on the FeSiAl flakes along with HC after the composite is formed and also the ZnO to prove the claims made on the process in syntheses. 

  1. TEM analysis can be performed also to understand the materials involved 

  1. XPS analysis should be performed to check the different metal species in the composite 

  1. TGA of all the samples should be shown to understand the comparison between the different samples

Author Response

Dear reviewer, 

Thank you very much for your valuable comments.

Kindness regards

Round 2

Reviewer 2 Report

The manuscript can be accepted.